# THERML:
## THE THERMODYNAMICS OF MACHINE LEARNING

### ABSTRACT

In this work we offer an information-theoretic framework for representation learning that connects with a wide class of existing objectives in machine learning. We develop a formal correspondence between this work and thermodynamics and discuss its implications.

## 1 INTRODUCTION

Let $X, Y$ be some paired data, for example: a set of images $X$ and their labels $Y$. We imagine the data comes from some *true*, unknown data generating process $\Phi$[1], from which we have drawn a *training set* of $N$ pairs:

$$\mathcal{T}_N \equiv (x^N, y^N) \equiv \{x_1, y_1, x_2, y_2, \ldots, x_N, y_N\} \sim \phi(x^N, y^N). \tag{1}$$

We further imagine the process is *exchangeable*[2] and the data is conditionally independent given the governing process $\Phi$:

$$p(x^N, y^N | \phi) = \prod_i p(x_i | \phi) p(y_i | x_i, \phi). \tag{2}$$

As machine learners, we believe that by studying the training set, we should be able to infer or predict new draws from the same data generating process. Call a set of $M$ future draws from the data generating process $\mathcal{T}'_M \equiv \{X^M, Y^M\}$ the *test set*.

The *predictive information* (Bialek et al., 2001) is the *mutual information* between the training set and a infinite test set, equivalently the amount of information the training set provides about the generative process itself:

$$I_{\text{pred}}(\mathcal{T}_N) \equiv \lim_{M \to \infty} I(\mathcal{T}_N; \mathcal{T}'_M) = I(\mathcal{T}_N; \Phi) = I(X^N, Y^N; \Phi). \tag{3}$$

The predictive information measures the underlying complexity of the data generating process (Still, 2014), and is fundamentally limited and must grow sublinearly in the dataset size (Bialek et al., 2001). Hence, the predictive information is a vanishing fraction of the total information in the training set [3]:

$$\lim_{N \to \infty} \frac{I_{\text{pred}}(\mathcal{T}_N)}{H(\mathcal{T}_N)} = 0 \tag{4}$$

A vanishing fraction of the information present in our training data is in any way useful for future tasks. A vanishing fraction of the information contained in the training data is *signal*, the rest is *noise*. We claim the goal of learning is to learn a *representation* of data, both locally and globally that captures the predictive information while being maximally compressed: that separates the signal from the noise.

---

[1] Here we aim to invoke the same philosophy as in the introduction to Watanabe (2018).

[2] That is, we imagine the data satisfies De Finetti's theorem, for which infinite exchangeable processes usually can be described by products of conditionally independent distributions, but don't want to worry too much about the complicated details since there are subtle special cases (Accardi, 2018).

[3] Here and throughout $H(A)$ is used to denote entropies $H(A) = -\sum_i p(A) \log p(A)$.

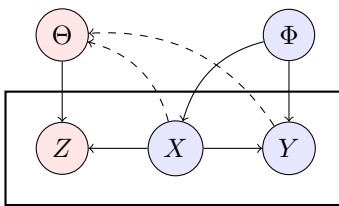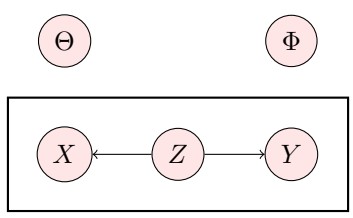

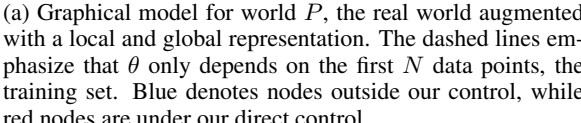

(a) Graphical model for world $P$, the real world augmented with a local and global representation. The dashed lines emphasize that $\theta$ only depends on the first $N$ data points, the training set. Blue denotes nodes outside our control, while red nodes are under our direct control.

(b) Graphical model for world $Q$, the world we desire. In this world, $Z$ acts as a latent variable for $X$ and $Y$ jointly.

Figure 1: Graphical models.

## 2 A Tale of Two Worlds

We are primarily interested in learning a stochastic local *representation* of $X$, call it $Z$, defined by some parametric distribution of our own design: $p(z_i|x_i, \theta)$ with its own parameters $\theta$. A *training procedure* is a process that assigns a distribution $p(\theta|x^N, y^N)$ to the parameters conditioned on the observed dataset. In this way, the parameters of our local parametric map are themselves a global representation of the dataset. With our augmentations, the world now looks like the graphical model in Figure 3a, denoted World $P$: Some data generating process $\Phi$ generates a dataset $(X^N, Y^N)$ which we perform some learning algorithm on to get some parameters $p(\theta|x^N, y^N)$ which we can use to form a parametric local representation $p(z_i|x_i, \theta)$.

World $P$ is what we have. It is not necessarily what we want. What we *have* to contend with is an unknown distribution of our data. What we *want* is a world that corresponds to the traditional modeling assumptions in which $Z$ acts as a latent factor for $X$ and $Y$, rendering them conditionally independent, leaving no correlations unexplained. Similarly, we would prefer if we could easily marginalize out the dependence on our universal ($\Phi$) and model specific ($\Theta$) parameters. World $Q$ in Figure 3b is the world we *want* [4].

We can measure the degree to which the real world aligns with our desires by computing the minimum possible *relative information*[5] between our distribution $p$ and any distribution consistent with the conditional dependencies encoded in graphical model $Q$[6]. It can be shown (Friedman et al., 2001) that this quantity is given by the difference in multi-informations between the two graphical models, as measured in World $P$:

$$\mathcal{J} \equiv \min_{q \in Q} D_{\mathrm{KL}}\left[p; q\right] = I_P - I_Q. \tag{5}$$

The *multi-information* (Slonim et al., 2005) of a graphical model is the KL divergence between the joint distribution and the product of all of the marginal distributions, which can be computed as a sum of mutual informations, one for each node in the graph, between itself and its parents:

$$I_G \equiv \left\langle \log \frac{p(g^N)}{\prod_i p(g_i)} \right\rangle = \sum_i I(g_i; \mathrm{Pa}(g_i)) \tag{6}$$

In our case:

$$\mathcal{J} = I(\Theta; X^N, Y^N) + \sum_i \left[ I(X_i; \Phi) + I(Y_i; X_i, \Phi) + I(Z_i; X_i, \Theta) - I(X_i; Z_i) - I(Y_i; Z_i) \right]. \tag{7}$$

---

[4] We could consider different alternatives, deciding to relax some of the constraints we imposed in World $Q$, or generalizing World $P$ by letting the representation depend on $X$ and $Y$ jointly, for instance. What follows demonstrates a general sort of *calculus* that we can invoke for any specified pair of graphical models. In particular Appendices A to C discuss alternatives.

[5] Also known as the KL divergence.

[6] Note that this is $D_{\mathrm{KL}}\left[p; q^*\right]$ where $q^*$ is the well known reverse-*information projection* or *moment projection*: $q^* = \mathrm{argmin}_{q \in Q} D_{\mathrm{KL}}\left[p; q\right]$ (Csiszár & Matúš, 2003).

This minimal relative information has two terms outside our control and we can take them to be constant, but which relate to the predictive information:

$$\sum_i \left[ I(X_i; \Phi) + I(Y_i; X_i, \Phi) \right] \geq \sum_i I(Y_i; X_i) + I_{\text{pred}}(\mathcal{T}_N). \tag{8}$$

These terms measure the intrinsic complexity of our data. The remaining four terms are:

- $I(X_i; Z_i)$ - which measures how much information our representation contains about the input $(X)$. This should be maximized to ensure our local representation actually represents the input.

- $I(Y_i; Z_i)$ - which measures how much information our representation contains about our auxiliary data. This should be maximized as well to ensure that our local representation is predictive for the labels.

- $I(Z_i; X_i, \Theta)$ - which measures how much information the parameters and input determine about our representation. This should be minimized to ensure consistency between worlds, and ensure we learn compressed local representations. Notice that this is similar to, but distinct from the first term above.

$$I(Z_i; X_i, \Theta) = I(Z_i; X_i) + I(Z_i; \Theta | X_i) \tag{9}$$

  by the Chain Rule for mutual information [7].

- $I(\Theta; X^N, Y^N)$ - which measures how much information we store about our training data in the parameters of our encoder. This should also be minimized to ensure we learn compressed global representation, preveting overfitting.

These mutual informations are all intractable in general, since we cannot compute the necessary marginals in closed form, given that we do not have access to the true data generating distribution.

## 2.1 FUNCTIONALS

Despite their intractability, we can compute variational bounds on these mutual informations.

### 2.1.1 ENTROPY

$$S \equiv \left\langle \log \frac{p(\theta | x^N, y^N)}{q(\theta)} \right\rangle_P \geq I(\Theta; X^N, Y^N) \tag{10}$$

The relative entropy in our parameters or just *entropy* for short measures the relative information between the distribution we assign our parameters in World $P$ after learning from the data $(X^N, Y^N)$, with respect to some data independent $q(\theta)$ *prior* on the parameters. This is an upper bound on the mutual information between the data and our parameters and as such can measure our risk of overfitting our parameters.

### 2.1.2 RATE

$$R_i \equiv \left\langle \log \frac{p(z_i | x_i, \theta)}{q(z_i)} \right\rangle_P \geq I(Z_i; X_i, \Theta) \tag{11}$$

The *rate* measures the complexity of our representation. It is the relative information of a sample specific representation $z_i \sim p(z|x_i, \theta)$ with respect to our variational marginal $q(z)$. It measures how many bits we actually encode about each sample, and can measure how our risk of overfitting our representation. We use $R \equiv \sum_i R_i$.

---

[7]Given this relationship, we could actually reduce the total number of functions we consider from 4 to 3, as discussed in Appendix A.

### 2.1.3 CLASSIFICATION ERROR

$$C_i \equiv - \langle \log q(y_i|z_i) \rangle_P \geq H(Y_i) - I(Y_i; Z_i) = H(Y_i|Z_i) \tag{12}$$

The *classification error* measures the conditional entropy of $Y$ left after conditioning on $Z$. It is a measure of how much information about $Y$ is left unspecified in our representation. This functional measures our supervised learning performance. We use $C \equiv \sum_i C_i$.

### 2.1.4 DISTORTION

$$D_i \equiv - \langle \log q(x_i|z_i) \rangle_P \geq H(X_i) - I(X_i; Z_i) = H(X_i|Z_i) \tag{13}$$

The *distortion* measures the conditional entropy of $X$ left after conditioning on $Z$. It is a measure of how much information about $X$ is left unspecified in our representation. This functional measures our unsupervised learning performance. We use $D \equiv \sum_i D_i$.

## 2.2 GEOMETRY

The distributions $p(z|x, \theta), p(\theta|x^N, y^N), q(z), q(x|z), q(y|z)$ can be chosen arbitrarily. Once chosen, the *functionals* $R, C, D, S$ take on well described values. The choice of the five distributional families specifies a single point in a four-dimensional space.

Importantly, the sum of these functionals is a variational upper bound (up to an additive constant) for the minimum possible relative information between worlds (Appendix D):

$$S + R + C + D \geq \mathcal{J} + \sum_i H(X_i, Y_i|\Phi) \tag{14}$$

Besides just the upper bound, we can consider the full space of *feasible* points. Notice that $S$ and $R$ are both themselves upper bounds on mutual informations, and so must be positive semi-definite. If our data is discrete, or if we have discretized it [8], $D$ and $C$ which are both upper bounds on conditional entropies, must be positive as well. Along with Equation (14), given that $\sum_i H(X_i, Y_i|\Phi)$ is a positive constant outside our control, the space of possible $(R, C, D, S)$ values is at least restricted to be points in the positive orthant with some minimum possible Manhattan distance to the origin:

$$S + R + C + D \geq \sum_i H(X_i, Y_i|\Phi) \qquad R \geq 0 \quad S \geq 0 \quad D \geq 0 \quad C \geq 0 \tag{15}$$

Even in the infinite model family limit, data-processing inequalities on mutual information terms all defined in a set of variables that satisfy some nontrivial conditional dependencies ensure that there are regions in this functional space that are wholly out of reach. The surface of the feasible region maps an optimal frontier, optimal in the degree to which it minimizes mismatch between our two worlds subject to constraints on the relative magnitudes of the individual terms. This convex polytope has edges, faces and corners that are identifiable as the optimal solutions for well known objectives.

This story is a generalization of the story presented in Alemi et al. (2018), which can be considered a two-dimensional projection of this larger space (onto $R, D$). Within our larger framework we can derive more specific bounds between subsets of the functionals. For instance:

$$R_i + D_i \geq H(X_i) + I(Z_i; \Theta|X_i). \tag{16}$$

This mirrors the bound given in Alemi et al. (2018) where $R + D \geq H(X)$, which is still true given that all conditional mutual informations are positive semi-definite $(H(X) + I(Z; \Theta|X) \geq H(X))$, but here we obtain a tighter pointwise bound that has a term measuring how much information about our encoding is revealed by the parameters after conditioning on the input itself. This term

---

[8]More generally, if we choose some measure $m(x), m(y)$ on both $X$ and $Y$, we can define $D$ and $C$ in terms of that measure e.g. $D \equiv - \left\langle \log \frac{q(x|z)}{m(x)} \right\rangle_P \geq H_m(X) - I(X; Z) = H_m(X|Z)$

$I(Z_i; \Theta | X_i)$ captures the degree to which our local representation is overly sensitive to the particular parameter settings [9][10].

## 2.3 GENERALIZATION

We can evaluate how much information our representations capture about the true data generating process. For instance, $I(Z_i; \Phi)$ which measures how much information about the true data generating procedure our local representations capture. Notice that given the conditional dependencies in world $P$, we have the following Markov chain:

$$\Phi \to (X_i, Y_i, \Theta) \to Z_i \tag{17}$$

and so by the Data Processing Inequality (Cover & Thomas, 2012):

$$I(Z_i; \Phi) \leq I(Z_i; \Theta, X_i, Y_i) = I(Z_i; X_i, \Theta) + \underline{I(Z_i; Y_i | X_i, \Theta)} \leq R_i. \tag{18}$$

The per-instance rate $R_i$ forms an upper bound on the mutual information between our encoding $Z_i$ and the *true* governing parameters of our data $\Phi$. Similarly, we can establish that:

$$\Phi \to (X^N, Y^N) \to \Theta \implies I(\Theta; \Phi) \leq I(\Theta; X^N, Y^N) \leq S. \tag{19}$$

$S$ upper bounds the amount of information our encoder's parameters $\Theta$, the global representation of the dataset can contain about the true process $\Phi$. At the same time:

$$I(\Theta; \Phi) \leq I(X^N, Y^N; \Phi) \leq \sum_i I(X_i, Y_i; \Phi), \tag{20}$$

which sets a natural upper limit for the maximum $S$ that might be useful.

## 3 OPTIMAL FRONTIER

As in Alemi et al. (2018), under mild assumptions about the variational distributional families, it can be argued that the surface is monotonic in all of its arguments. The optimal surface in the infinite family limit can be characterized as a convex polytope (Equation (15)). In practice we will be in the realistic setting corresponding to finite parametric families such as neural network approximators. We then expect that there is an irrevocable gap that opens up in the variational bounds. Any failure of the distributional families to model the correct corresponding marginal in $P$ means that the space of all *realizable* $R, C, D, S$ values will be some convex relaxation of the optimal *feasible* surface. This surface will be described some function $f(R, C, D, S) = 0$, which means we can identify points on the surface as a function of one functional with respect to the others (e.g. $R = R(C, D, S)$). Finding points on this surface equates to solving a constrained optimization problem, e.g.

$$\min_{q(z)q(x|z)q(y|z)p(z|x,\theta)p(\theta|\{x,y\})} R \text{ such that } D = D_0, S = S_0, C = C_0. \tag{21}$$

Equivalently, we could solve the unconstrained Lagrange multipliers problem:

$$\min_{q(z)q(x|z)q(y|z)p(z|x,\theta)p(\theta|\{x,y\})} R + \delta D + \gamma C + \sigma S. \tag{22}$$

Here $\delta, \gamma, \sigma$ are Lagrange multipliers that impose the constraints. They each correspond to the partial derivative of the rate at the solution with respect to their corresponding functional, keeping the others fixed.

Notice that this single objective encompasses a wide range of existing techniques.

- If we retain $C$ alone, we are doing traditional supervised learning and our network will learn to be deterministic in its activations and parameters.

---

[9] In Appendix A we consider taking this bound seriously to limit the space only only three functionals, $S$, $C$ and $V \geq I(Z_i; \Theta | X_i)$

[10] This could help explain the observation that often times putting additional modeling power on the prior rather than the encoder can give improvements in ELBO (Chen et al., 2016).

- If $\delta = 0$ we no longer require a variational reconstruction network $q(x|z)$, and are doing some form of supervised learning generally.

- If $\delta = 0, \sigma = 0$ we exactly recover the Variational Information Bottleneck (VIB) objective of Alemi et al. (2016) (where $\beta = 1/\gamma$), a form of stochastically regularized supervised learning that imposes a bottleneck on how much information our representation can retain about the input, while simultaneously maximizing the amount of information the representation contains about the target.

- If $\delta = 0$ and $\sigma, \gamma \to \infty$ but in such a way as to keep the ratio fixed $\beta \equiv \sigma/\gamma$ (that is if we drop the $R$ term and only keep $C + \beta S$ as our objective) we recover the Information Bottleneck Lagrangian loss of Achille & Soatto (2017), presented as an alternative way to do Information Bottleneck (Tishby et al., 1999) but being stochastic on the parameters rather than the activations as in VIB.

- As a special case, if our objective is set to $C + S$ ($\delta = 0, \sigma, \gamma \to \infty, \sigma/\gamma \to 1$), we obtain the objective for a Bayesian neural network, ala Blundell et al. (2015).

- If we retain only $D$, we are training a stochastic autoencoder.

- If $\sigma = 0, \gamma = 0, \delta = 1$ the objective is equivalent to the ELBO used to train a VAE (Kingma & Welling, 2014).

- If $\sigma = 0, \gamma = 0$ more generally, the objective is equivalent to a $\beta$-VAE (Higgins et al., 2017) where $\beta = 1/\delta$.

- If $\gamma = 0$ all terms involving the auxiliary data $Y$ drop out and we are doing some form of unsupervised learning without any variational classifier $q(y|z)$. The presence of the $S$ term makes this more general than a usual $\beta$-VAE and should offer better generalization properties and control of overfitting by bottle-necking how much information we allow the parameters of our encoder to extract from the training data.

- $\sigma = 0, \gamma = \alpha, \delta = 1$ recovers the semi-supervised objective of Kingma et al. (2014).

- In its most general form, in common parlance the full objective might be described as a temperature-regulated Bayesian semi-supervised $\beta$-VAE, or a Variational Information Bottleneck Lagrangian Autoencoder (VIBLA).

Examples of all of these objectives behavior on a simple toy model is shown in Appendix H.

Notice that all of these previous approaches describe low dimensional sub-surfaces of the optimal three dimensional frontier. These approaches were all interested in different domains, some were focused on supervised prediction accuracy, others on learning a generative model. Depending on your specific problem, and downstream tasks, different points on the optimal frontier will be desirable. However, instead of choosing a single point on the frontier, we can now explore a region on the surface to see what class of solutions are possible within the modeling choices. By simply adjusting the three control parameters $\delta, \gamma, \sigma$, we can smoothly move across the entire frontier and smoothly interpolate between all of these objectives and beyond.

### 3.1 OPTIMIZATION

So far we've considered explicit forms of the objective in terms of the four functionals. For $S$ this would require some kind of tractable approximation to the posterior over the parameters of our encoding distribution[11]. Alternatively, we can formally describe the exact solution to our minimization problem:

$$\min S \text{ s.t. } R = R_0, C = C_0, D = D_0. \tag{23}$$

Recall that $S$ measures the relative entropy of our parameter distribution with respect to the $q(\theta)$ *prior*. As such, the solution that minimizes the relative entropy subject to some constraints is a generalized Boltzmann distribution (Jaynes, 1957):

$$p^*(\theta|\{x, y\}) = \frac{q(\theta)}{\mathcal{Z}} e^{-(R + \delta D + \gamma C)/\sigma}. \tag{24}$$

---

[11] As in Blundell et al. (2015); Achille & Soatto (2017)

Here $\mathcal{Z}$ is the *partition function*, the normalization constant for the distribution

$$\mathcal{Z} = \int d\theta \, q(\theta) \, e^{-(R+\delta D+\gamma C)/\sigma} \tag{25}$$

This suggests an alternative method for finding points on the optimal frontier. We could turn the unconstrained Lagrange optimization problem that required some explicit choice of tractable posterior distribution over parameters into a sampling problem for a richer implicit distribution.

A naive way to draw samples from this posterior would be to use Stochastic Gradient Langevin Dynamics or its cousins (Welling & Teh, 2011; Chen et al., 2014; Ma et al., 2015) which, in practice, would look like ordinary stochastic gradient descent (or its cousins like momentum) for the objective $R + \delta D + \gamma C$, with injected noise. By choosing the magnitude of the noise relative to the learning rate, the effective temperature $\sigma$ can be controlled.

There is increasing evidence that the stochastic part of stochastic gradient descent itself is enough to turn SGD less into an optimization procedure and more into an approximate posterior sampler (Mandt et al., 2017; Smith & Le, 2017; Achille & Soatto, 2017; Zhang et al., 2018; Chaudhari & Soatto, 2017), where hyperparameters such as the learning rate and batch size set the effective temperature. If ordinary stochastic gradient descent is doing something more akin to sampling from a posterior and less like optimizing to some minimum, it would help explain improved performance through ensemble averages of different points along trajectories (Huang et al., 2017).

When viewed in this light, Equation 24 describes the optimal posterior for the parameters so as to ensure the minimal divergence between worlds $P$ and $Q$. $q(\theta)$ plays the role of the *prior* over parameters, but our overall objective is minimized when

$$q(\theta) = p(\theta) = \langle p(\theta|x^N, y^N)\rangle_{p(x^N, y^N)}. \tag{26}$$

That is, when our *prior* is the marginal of the posteriors over all possible datasets drawn from the true distribution. A fair draw from this marginal is to take a sample from the posterior obtained on a different but related dataset. Insomuch as ordinary SGD training is an approximate method for drawing a posterior sample, the common practice of fine-tuning a pretrained network on a related dataset is using a sample from the optimal *prior* as our initial parameters. The fact that fine-tuning approximates use of an optimal *prior* presumably helps explain its broad success.

If we identify our true goal not as optimizing some objective but instead directly sampling from Equation 24, we can consider alternative approaches to define our learning dynamics, such as *parallel tempering* or *population annealing* (Machta & Ellis, 2011). Alternatively, we could, instead of adopting variational bounds on the mutual informations, consider other mutual information bounds such as those in Ishmael Belghazi et al. (2018); van den Oord et al. (2018). Perhaps our priors can be fit, providing we form estimates of the expectation over datasets (e.g. bootstrapping or jackknifing our dataset (DasGupta, 2008)).

## 4 THERMODYNAMICS

So far we have described a framework for learning that involves finding points that lie on the surface of a convex three-dimensional surface in terms of four functional coordinates $R, C, D, S$. Interestingly, this is all that is required to establish a formal connection to thermodynamics, which similarly is little more than the study of exact differentials (Sethna, 2006; Finn, 1993).

Whereas previous approaches connecting thermodynamics and learning (Parrondo et al., 2015; Still, 2017; Still et al., 2012) have focused on describing the thermodynamics and statistical mechanics of physical realizations of learning systems (i.e. the heat bath in these papers is a physical heat bath at finite temperature), in this work we make a formal analogy to the structure of the theory of thermodynamics, without any physical content.

### 4.1 FIRST LAW OF LEARNING

The optimal frontier creates an equivalence class of states, being the set of all states that minimize as much as possible the distortion introduced in projecting world $P$ onto a set of distributions that

respect the conditions in $Q$. The surface satisfies some equation $f(R, C, D, S) = 0$ which we can use to describe any one of these functionals in terms of the rest, e.g. $R = R(C, D, S)$. This function is entire, and so we can equate partial derivatives of the function with differentials of the functionals[12]:

$$dR = \left(\frac{\partial R}{\partial C}\right)_{D,S} dC + \left(\frac{\partial R}{\partial D}\right)_{C,S} dD + \left(\frac{\partial R}{\partial S}\right)_{C,D} dS. \tag{27}$$

Since the function is smooth and convex, instead of identifying the surface of optimal rates in terms of the functionals $C, D, S$, we could just as well describe the surface in terms of the partial derivatives by applying a Legendre transformation. We will name the partial derivatives:

$$\gamma \equiv -\left(\frac{\partial R}{\partial C}\right)_{D,S} \qquad \delta \equiv -\left(\frac{\partial R}{\partial D}\right)_{C,S} \qquad \sigma \equiv -\left(\frac{\partial R}{\partial S}\right)_{C,D}. \tag{28}$$

These measure the exchange rate for turning rate into reduced distortion, reduced classification error, or increased entropy, respectively.

The functionals $R, C, D, S$ are analogous to extensive thermodynamic variables such as volume, entropy, particle number, magnetic field, charge, surface area, length and energy which grow as the system grows, while the named partial derivatives $\gamma, \delta, \sigma$ are analogous to the intensive, generalized forces in thermodynamics corresponding to their paired state variable, such as pressure, temperature, chemical potential, magnetization, electromotive force, surface tension, elastic force, etc. Just as in thermodynamics, the *extensive* functionals are defined for any state, while the *intensive* partial derivatives are only well defined for *equilibrium states*, which in our language are the states lying on the optimal surface [13].

Recasting our total differential:

$$dR = -\gamma dC - \delta dD - \sigma dS, \tag{29}$$

we create a law analogous to the *First Law of Thermodynamics*. In thermodynamics the First Law is often taken to be a statement about the conservation of energy, and by analogy here we could think about this *law* as a statement about the conservation of information. Granted, the actual content of the law is fairly vacuous, equivalent only to the statement that there exists a scalar function $R = R(C, D, S)$ defining our surface and its partial derivatives.

## 4.2 MAXWELL RELATIONS AND THERMODYNAMIC POTENTIALS

Requiring that Equation 29 be an exact differential has mathematically trivial but intuitively non-obvious implications that relate various partial derivatives of the system to one another, akin to the *Maxwell Relations* in thermodynamics. For example, requiring that mixed second partial derivatives are symmetric establishes that:

$$\left(\frac{\partial^2 R}{\partial D \partial C}\right) = \left(\frac{\partial^2 R}{\partial C \partial D}\right) \implies \left(\frac{\partial \delta}{\partial C}\right)_D = \left(\frac{\partial \gamma}{\partial D}\right)_C. \tag{30}$$

This equates the result of two very different experiments. In the experiment encoded in the partial derivative on the left, one would measure the change in the derivative of the $R - D$ curve ($\delta$) as a function of the classification error ($C$) at fixed distortion ($D$). On the right one would measure the change in the derivative of the $R - C$ curve ($\gamma$) as a function of the distortion ($D$) at fixed classification error ($C$). As different as these scenarios appear, they are mathematically equivalent. A full set of Maxwell relations can be found in Appendix F.

We can additionally take and name higher order partial derivatives, analogous to the susceptibilities of thermodynamics like bulk modulus, the thermal expansion coefficient, or heat capacities. For instance, we can define the analog of heat capacity for our system, a sort of rate capacity at constant distortion:

$$K_D \equiv \left(\frac{\partial R}{\partial \sigma}\right)_D. \tag{31}$$

---

[12] $\left(\frac{\partial X}{\partial Y}\right)_Z$ denotes the partial derivative of $X$ with respect to $Y$ holding $Z$ constant.

[13] For more discussion of equilibrium states, and how they connect with more intuitive notions of equilibrium, see Appendix G

Just as in thermodynamics, these susceptibilities may offer useful ways to characterize and quantify the systematic differences between model families. Perhaps general scaling laws can be found between susceptibilities and network widths, or depths, or number of parameters or dataset size. Divergences or discontinuities in the susceptibilities are the hallmark of phase transitions in physical systems, and it is reasonable to expect to see similar phenomenon for certain models.

A great deal of first, second and third order partial derivatives in thermodynamics are given unique names. This is because the quantities are particularly useful for comparing different physical systems. We expect a subset of the first, second and higher order partial derivatives of the base functionals will prove similarly useful for comparing, quantifying, and understanding differences between modeling choices.

### 4.3 SECOND LAW OF LEARNING?

Even when doing deterministic training, training is non-invertible (Maclaurin et al., 2015), and we need to contend with and track the entropy ($S$) term. We set the parameters of our networks initially with a fair draw from some prior distribution $q(\theta)$. The training procedure acts as a Markov process on the distribution of parameters, transforming it from the prior distribution into some modified distribution, the posterior $p(\theta|x^N, y^N)$. Optimization is a many-to-one function, that in the ideal limiting case, maps all possible initializations to a single global optimum. In this limiting case $S$ would be divergent, and there is nothing to prevent us from memorizing the training set.

The Second Law of Thermodynamics states that the entropy of an isolated system tends to increase. All systems tend to disorder, and this places limits on the maximum possible efficiency of heat engines.

Formally, there are many statements akin to the Second Law of Thermodynamics that can be made about Markov chains generally (Cover & Thomas, 2012). The central one is that for any for any two distributions $p_n, q_n$ both evolving according to the same Markov process ($n$ marks the time step), the relative entropy $D_{\mathrm{KL}}[p_n; q_n]$ is monotonically decreasing with time. This establishes that for a stationary Markov chain, the relative entropy to the stationary state $D_{\mathrm{KL}}[p_n; p_\infty]$ monotonically decreases [14].

In our language, we can make strong statements about dynamics that target points on the optimal frontier, or dynamics that implement a relaxation towards equilibrium. There is a fundamental distinction between states that live on the frontier and those off of it, analogous to the distinction between equilibrium and non-equilibrium states in thermodynamics.

Any equilibrium distribution can be expressed in the form Equation (24) and identified by its partial derivatives $\gamma, \delta, \sigma$. If name the objective in Equation (22):

$$J(\gamma, \delta, \sigma) \equiv R + \delta D + \gamma C + \sigma S, \tag{32}$$

The value this objective takes for any equilibrium distribution can be shown to be given by the log partition function (Equation (25)):

$$\min J(\gamma, \delta, \sigma) = -\sigma \log \mathcal{Z}(\gamma, \delta, \sigma) \tag{33}$$

and the KL divergence between any distribution over parameters $p(\theta)$ and an equilibrium distribution is:

$$D_{\mathrm{KL}}[p(\theta); p^*(\theta; \gamma, \delta, \sigma)] = \Delta J/\sigma \tag{34}$$

$$\Delta J \equiv J^{\mathrm{noneq}}(p; \gamma, \delta, \sigma) - J(\gamma, \delta, \sigma) \tag{35}$$

Where $J^{\mathrm{noneq}}$ is the non-equilibrium objective:

$$J^{\mathrm{noneq}}(p; \gamma, \delta, \sigma) = \langle R + \delta D + \gamma C + \sigma S \rangle_{p(\theta)}. \tag{36}$$

For a stationary Markov process whose stationary distribution is an equilibrium distribution the KL divergence to the stationary distribution must monotonically decrease each step. This means the $\Delta J/\sigma$ must decrease monotonically, that is our objective $J$ must decrease monotonically:

$$J_{t=0} \geq J_t \geq J_{t+1} \geq J_{t=\infty}. \tag{37}$$

---

[14]For discrete state Markov chains, this implies that if the stationary distribution is uniform, the entropy of the distribution $H(p_n)$ is strictly increasing.

Furthermore, if we use $q(\theta)$ as our prior over parameters, we know:

$$J_{t=0} = \langle R + \delta D + \gamma C \rangle_{q(\theta)} \tag{38}$$

$$J_{t=\infty} = -\sigma \log Z. \tag{39}$$

## 5 CONCLUSION

We have formalized representation learning as the process of minimizing the distortion introduced when we project the real world (World $P$) onto the world we desire (World $Q$). The projection is naturally described by a set of four functionals which variationally bound relevant mutual informations in the real world. Relations between the functionals describe an optimal three-dimensional surface in a four dimensional space of *optimal* states. A single learning objective targeting points on this optimal surface can express a wide array of existing learning objectives spanning from unsupervised learning to supervised learning and everywhere in between. The geometry of the optimal frontier suggests a wide array of identities involving the functionals and their partial derivatives. This offers a direct analogy to thermodynamics independent of any physical content. By analogy to thermodynamics, we can begin to develop new quantitative measures and relationships amongst properties of our models that we believe will offer a new class of theoretical understanding of learning behavior.

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

## A    RECONSTRUCTION FREE FORMULATION

We can utilize the Chain Rule of Mutual Information (Equation (9)):

$$I(Z_i; X_i, \Theta) = I(Z_i; X_i) + I(Z_i; \Theta|X_i), \tag{40}$$

to simplify our expression for the minimum possible KL between worlds (Equation (7)), and consider a reduced set of functionals (compare to Section 2.1):

- $C_i \equiv - \langle \log q(y_i|z_i) \rangle_P \geq H(Y_i) - I(Y_i; Z_i) = H(Y_i|Z_i)$
  The *classification error*, as before.

- $S \equiv \left\langle \log \frac{p(\theta|\{x,y\})}{q(\theta)} \right\rangle_P \geq I(\Theta; \{X, Y\})$
  The *entropy* as before.

- $V_i \equiv \left\langle \log \frac{p(z_i|x_i,\theta)}{q(z_i|x_i)} \right\rangle_P \geq I(Z_i; \Theta|X_i)$
  The *volume* of the representation (for lack of a better term), which measures the mutual information between our representation $Z$ and the parameters $\Theta$, conditioned on the input $X$. That is, this functional bounds how much of the information in our representation can come from the learning algorithm, independent of the actual input.

In principle, these three functionals still fully characterize the distortion introduced in our information projection. Notice that this new functional requires the variational approximation $q(z_i|x_i)$, a variational approximation to the marginal over our parameter distribution. Notice also that we no longer require a variational approximation to $p(x_i|z_i)$. That is, in this formulation we no longer require any form of decoder, or synthesis in our original data space $X$. While equivalent in its information projection, this more naturally corresponds to the model of our desired world $Q$:

$$q(x, y, \phi, z, \theta) = q(\phi)q(\theta) \prod_i q(z_i|x_i)q(y_i|z_i), \tag{41}$$

depicted below in Figure 2. Here we desire, not the joint generative model $X \leftarrow Z \rightarrow Y$, but the predictive model $X \rightarrow Z \rightarrow Y$.

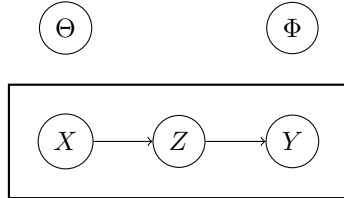

Figure 2: Modified graphical model for world $Q$, instead of Figure 3b, the world we desire which satisfies the joint density in Equation 41. Notice that this graphical model encodes all of the same conditional independencies as the original.

In this case we have:

$$C + S + V \geq \mathcal{J} + \sum_i \left[ H(Y_i X_i|\Phi) - H(X_i) \right]. \tag{42}$$

We can imagine tracing out this, now three dimensional, frontier that still explores a space consistent with our original graphical model, but wherein we no longer have to do any form of direct variational synthesis.

## B    BAYESIAN INFERENCE

Just as in A we can consider alternative graphical models for World P. In particular, we can consider a simplified scenario depicted in Figure 3 corresponding to the usual situation in Bayesian inference.

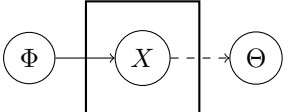
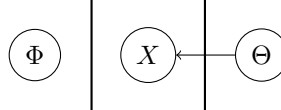

(a) Graphical model for world $P$, depicting Bayesian inference as learning a single global representation of data.

(b) Graphical model for world $Q$, the world we desire, the usual generative model of Bayesian inference.

Figure 3: Graphical models for standard Bayesian inference.

Here we have just data, generated by some process and we form a single global representation of the dataset. The world we desire, World Q, corresponds to the usual Bayesian modeling assumption, whereby our own global representation generates the data conditionally independently.

For these sets of graphical models, we have the following information projection:

$$\mathcal{J}_{\text{bayes}} = \min_{q \in Q} D_{\text{KL}}\left[p; q\right] = I_P - I_Q = \sum_i I(X_i; \Phi) + I(\Theta; X^n) - \sum_i I(X_i; \Theta) \tag{43}$$

And we can derive the simple variational bounds:

$$S \equiv \left\langle \log \frac{p(\theta|X^n)}{q(\theta)} \right\rangle \geq I(\Theta; X^n) \tag{44}$$

This *entropy* gives an upper bound on the mutual information between our parameters and the dataset, it requires a variational approximation to the true marginal of the posterior $p(\theta|X^N)$ over datasets: $q(\theta)$, a *prior*.

$$U_i \equiv -\left\langle \log q(x_i|\theta) \right\rangle \geq H(X_i|\Theta) \tag{45}$$

The *energy* gives an upper bound on the conditional entropy of our data given our parameters, it is powered by a variational approximation to the factored inverse of our global representation, the *likelihood* in ordinary parlance.

Our optimal frontier is set by those conditions above as well as: [15]:

$$U + S \geq \mathcal{J}_{\text{bayes}} + \sum_i H(X_i|\Phi) \tag{46}$$

Just as in our earlier paper (Alemi et al., 2018) we could trace out the frontier by doing the constrained optimization problem:

$$\min S + \beta U \tag{47}$$

The formal solution to this optimization problem takes the form:

$$\log p(\theta|x^N) = \log q(\theta) + \beta \sum_i \log q(x_i|\theta) - \log \mathcal{Z}. \tag{48}$$

Where $\mathcal{Z}$ is the partition function:

$$\mathcal{Z} = \int d\theta\, q(\theta) e^{\beta \sum_i \log q(x_i|\theta)} \tag{49}$$

This is the ordinary temperature regulated (Watanabe, 2009) Bayesian posterior:

$$p(\theta|x^N) \propto q(\theta) \prod_i q(x_i|\theta)^\beta. \tag{50}$$

Using a temperature to regulate the relative contribution of the prior and posterior has been used broadly, but ordinarily doesn't have a well founded justification. Here we can unapologetically vary

---

[15] $U \equiv \sum_i U_i$

the relative contributions of the prior and likelihood since in the representational framework, those are both variational approximations that might have differing ability to better model the true distributions they approximate. By varying the $\beta$ parameter here, just as in the $\beta$-VAE case (Alemi et al., 2018) we can smoothly explore the frontier within our modeling family, smoothly controlling the amount of information our model extracts from the dataset. This can help us control for overfitting in a principled way.

Additionally, we could try to relax our variational approximations, and *fit* our prior, assuming we could estimate an expectation over datasets. One way to do that is with a bootstrap or jackknife procedure (DasGupta, 2008).

## C  DISCRIMINATIVE MODELS

Similarly we could consider the situation depicting usual discriminative learning, depicted in Figure 4.

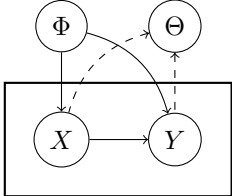

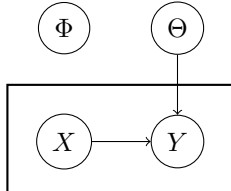

(a) Graphical model $P$, depicting conditionally independent data with a global representation.

(b) Graphic model $Q$ depicting a discriminative generative model.

Figure 4: Graphical models for the traditional discriminative case.

For these sets of graphical models, we have the following information projection:

$$\mathcal{J}_{\mathrm{d}} = I_P - I_Q = I(\Theta; X^N, Y^N) + \sum_i \left[ I(X_i; \Phi) + I(Y_i; X_i, \Phi) - I(Y_i; X_i, \Theta) \right]. \tag{51}$$

$$S \equiv \left\langle \log \frac{p(\theta|X^n)}{q(\theta)} \right\rangle_P \geq I(\Theta; X^n) \tag{52}$$

This *entropy* gives an upper bound on the mutual information between our parameters and the dataset, it requires a variational approximation to the true marginal of the posterior $p(\theta|X^N)$ over datasets: $q(\theta)$, a *prior*.

$$U_i \equiv - \left\langle \log q(y_i|x_i, \theta) \right\rangle_P \geq H(Y_i|X_i, \Theta) \tag{53}$$

The *energy* gives an upper bound on the conditional entropy of our targets given our parameters and input, it is powered by a variational approximation to the factored inverse of our global representation, the *conditional likelihood* in ordinary parlance.

Our optimal frontier is set by those conditions above as well as:

$$U + S \geq \mathcal{J}_{\mathrm{d}} + \sum_i H(Y_i|X_i, \Phi) - I(X_i; \Phi) \tag{54}$$

Just as previously in Appendix B solutions on the frontier can be specified by:

$$\log p(\theta|x^N, y^N) = \log q(\theta) + \beta \sum_i \log q(y_i|x_i, \theta) - \log \mathcal{Z}. \tag{55}$$

Here again we can smoothly explore the frontier set by the variationals approximations given by the prior and likelihood by simply adjusting $\beta$. We might additionally consider going beyond the fixed variational approximations and push the frontier by fitting the prior, or likelihood.

## D  FUNCTIONAL INEQUALITIES

Here we show the details for deriving Equation (14).

We start by expressing our functional inequalities, but being explicit about presence of the relative informations of our variational approximations.

$$I(\Theta; X^N, Y^N) = S - D_{\mathrm{KL}}\left[p(\theta); q(\theta)\right] \tag{56}$$

$$I(X_i; Z_i) = H(X_i) - D_i + D_{\mathrm{KL}}\left[p(x_i|z_i); q(x_i|z_i)\right] \tag{57}$$

$$I(Y_i; Z_i) = H(Y_i) - C_i + D_{\mathrm{KL}}\left[p(y_i|z_i); q(y_i|z_i)\right] \tag{58}$$

$$I(Z_i; X_i, \Theta) = R_i - D_{\mathrm{KL}}\left[p(z_i); q(z_i)\right] \tag{59}$$

Combining Equations (7) and (56) to (59):

$$\mathcal{J} = S + D + C + R - D_{\mathrm{KL}}\left[p; q\right] - \sum_i \left[H(X_i) + H(Y_i) - I(X_i; \Phi) - I(Y_i; X_i, \Phi)\right] \geq 0. \tag{60}$$

Here we have collected all of the KL divergences for our variational approximations:

$$\begin{aligned} D_{\mathrm{KL}}\left[p; q\right] \equiv & D_{\mathrm{KL}}\left[p(\theta); q(\theta)\right] + \sum_i D_{\mathrm{KL}}\left[p(x_i|z_i); q(x_i|z_i)\right] \\ & + \sum_i \left[D_{\mathrm{KL}}\left[p(y_i|z_i); q(y_i|z_i)\right] + D_{\mathrm{KL}}\left[p(z_i); q(z_i)\right]\right]. \end{aligned} \tag{61}$$

We can simplify:

$$H(X_i) - I(X_i; \Phi) = H(X_i|\Phi) \tag{62}$$

$$H(Y_i) - I(Y_i; X_i \Phi) = H(Y_i|X_i, \Phi) \tag{63}$$

$$H(Y_i|X_i, \Phi) + H(X_i|\Phi) = H(Y_i, X_i|\Phi) \tag{64}$$

To obtain:

$$\mathcal{J} = S + D + C + R - D_{\mathrm{KL}}\left[p; q\right] - \sum_i H(Y_i, X_i|\Phi) \tag{65}$$

Which yields:

$$S + D + C + R = \mathcal{J} + D_{\mathrm{KL}}\left[p; q\right] + \sum_i H(Y_i, X_i|\Phi) \tag{66}$$

$$S + D + C + R \geq \mathcal{J} + \sum_i H(Y_i, X_i|\Phi) \tag{67}$$

$$S + D + C + R \geq \sum_i H(Y_i, X_i|\Phi) \tag{68}$$

$$\tag{69}$$

## E  IDENTITIES

We will utilize some basic information identities, first by definition

$$I(A; B) = H(A) - H(A|B) \tag{70}$$

$$= H(B) - H(B|A) \tag{71}$$

$$= H(A) + H(B) - H(A, B) \tag{72}$$

$$= H(A, B) - H(A|B) - H(B|A) \tag{73}$$

By the chain rule of mutual information:

$$I(A, B; C) = I(A; C) + I(B; C|A) \geq 0 \tag{74}$$

Mutual informations, and conditional mutual informations are always positive:

$$I(A; B) \geq 0 \tag{75}$$

$$I(A; B|C) \geq 0 \tag{76}$$

We will also use the following rule for conditional entropies

$$H(B|A) = H(A, B) - H(A) \tag{77}$$

# F  MAXWELL RELATIONS

We can also define other potentials analogous to the alternative thermodynamic potentials such as enthalpy, free energy, and Gibb's free energy by performing partial Legendre transformations. For instance, we can define a *free rate*:

$$F(C, D, \sigma) \equiv R + \sigma S \tag{78}$$

$$dF = -\gamma dC - \delta dD + S d\sigma. \tag{79}$$

The free rate measures the rate of our system, not as a function of $S$ (something difficult to keep fixed), but in terms of $\sigma$, a parameter in our loss or optimal posterior.

The free rate gives rise to other Maxwell relations such as

$$\left(\frac{\partial S}{\partial C}\right)_\sigma = -\left(\frac{\partial \gamma}{\partial \sigma}\right)_C, \tag{80}$$

which equates how much each additional bit of entropy ($S$) buys you in terms of classification error ($C$) at fixed effective temperature ($\sigma$), to a seemingly very different experiment where you measure the change in the effective supervised tension ($\gamma$, the slope on the $R - C$ curve) versus effective temperature ($\sigma$) at a fixed classification error ($C$).

## F.1  COMPLETE ENUMERATION

Here we enumerate a complete set of Maxwell Relations. First if we write $R = R(D, C, S)$:

$$dR = -\gamma dC - \delta dD - \sigma dS$$

$$\left(\frac{\partial \gamma}{\partial D}\right)_C = \left(\frac{\partial \delta}{\partial C}\right)_D \tag{81}$$

$$\left(\frac{\partial \delta}{\partial S}\right)_D = \left(\frac{\partial \sigma}{\partial D}\right)_S \tag{82}$$

$$\left(\frac{\partial \gamma}{\partial S}\right)_C = \left(\frac{\partial \sigma}{\partial C}\right)_S \tag{83}$$

Next transforming to $F = R + \sigma S = F(D, C, \sigma)$

$$dF = -\gamma dC - \delta dD + S d\sigma$$

$$\left(\frac{\partial \gamma}{\partial \sigma}\right)_C = -\left(\frac{\partial S}{\partial C}\right)_\sigma \tag{84}$$

$$\left(\frac{\partial \delta}{\partial S}\right)_D = -\left(\frac{\partial S}{\partial D}\right)_\sigma \tag{85}$$

Next transforming to $H = R + \gamma C = H(D, \gamma, S)$

$$dH = C d\gamma - \delta D - \sigma dS \tag{86}$$

$$\left(\frac{\partial C}{\partial D}\right)_\gamma = -\left(\frac{\partial \delta}{\partial \gamma}\right)_D \tag{87}$$

$$\left(\frac{\partial C}{\partial S}\right)_\gamma = -\left(\frac{\partial \sigma}{\partial \gamma}\right)_S \tag{88}$$

Next transforming to $G = R + \sigma S + \gamma C = G(D, \gamma, \sigma)$
$$dG = Cd\gamma - \delta dD + Sd\sigma \tag{89}$$

$$\left(\frac{\partial C}{\partial \sigma}\right)_\gamma = \left(\frac{\partial S}{\partial \gamma}\right)_\sigma \tag{90}$$

Next transforming to $A = R + \delta D = A(\delta, C, S)$
$$dA = -\gamma dC + Dd\delta - \sigma dS \tag{91}$$

$$\left(\frac{\partial \gamma}{\partial \delta}\right)_C = -\left(\frac{\partial D}{\partial C}\right)_\delta \tag{92}$$

$$\left(\frac{\partial D}{\partial \sigma}\right)_\delta = -\left(\frac{\partial \sigma}{\partial \delta}\right)_S \tag{93}$$

Finally transforming to $B = R + \delta D + \sigma S = B(\delta, C, \sigma)$
$$dB = -\gamma dC + Dd\delta + Sd\sigma \tag{94}$$

$$\left(\frac{\partial \gamma}{\partial \sigma}\right)_C = -\left(\frac{\partial S}{\partial C}\right)_\sigma \tag{95}$$

$$\left(\frac{\partial S}{\partial \delta}\right)_\sigma = \left(\frac{\partial D}{\partial \sigma}\right)_\delta \tag{96}$$

## G   ZEROTH LAW OF LEARNING

A central concept in thermodynamics is a notion of equilibrium. The so called Zeroth Law of thermodynamics defines thermal equilibrium as a sort of reflexive property of systems (Finn, 1993). If system $A$ is in thermal equilibrium with system $C$, and system $B$ is separately in thermal equilibrium with system $C$, then system $A$ and $B$ are in thermal equilibrium with each other.

When any sub-part of a system is in thermal equilibrium with any other sub-part, the system is said to be an equilibrium state.

In our framework, the points on the optimal surface are analogous to the equilibrium states, for which we have well defined partial derivatives. We can demonstrate that this notion of equilibrium agrees with a more intuitive notion of equilibrium between coupled systems. Imagine we have two different models, characterized by their own set of distributions, Model $A$ is defined by $p_A(z|x, \theta), p_A(\theta, \{x, y\}), q_A(z)$, and model $B$ by $p_B(z|x, \theta), p_B(\theta, \{x, y\}), q_B(z)$. Both models will have their own value for each of the functionals: $R_A, S_A, D_A, C_A$ and $R_B, S_B, D_B, C_B$. Each model defines its own representation $Z_A, Z_B$. Now imagine coupling the models, by forming the joint representation $Z_C = (Z_A, Z_B)$ formed by concatenating the two representations together. Now the governing distributions over $Z$ are simply the product of the two model's distributions, e.g. $q_C(z_C) = q_A(z_A)q_B(z_B)$. Thus the rate $R_C$ and entropy $S_C$ for the combined model is the sum of the individual models: $R_C = R_A + R_B, S_C = S_A + S_B$.

Now imagine we sample new states for the combined system which are maximally entropic with the constraint that the combined rate stay constant:

$$\min S \text{ s.t. } R = R_C \implies p(\theta|\{x, y\}) = \frac{q(\theta)}{\mathcal{Z}} e^{-R/\sigma}. \tag{97}$$

For the expectation of the two rates to be unchanged after they have been coupled and evolved holding their total rate fixed, we must have,

$$-\frac{1}{\sigma}R_A - \frac{1}{\sigma_B}R_B = -\frac{1}{\sigma_C}R_C = -\frac{1}{\sigma_C}(R_A + R_B) \implies \sigma_A = \sigma_B = \sigma_C. \tag{98}$$

Therefore, we can see that $\sigma$, the effective temperature, allows us to identify whether two systems are in thermal equilibrium with one another. Just as in thermodynamics, if two systems at different temperatures are coupled, some transfer takes place.

# H   EXPERIMENTS

We show examples of models trained on a toy dataset for all of the different objectives we define above. The dataset has both an infinite data variant, where overfitting is not a problem, and a finite data variant, where overfitting can be clearly observed for both reconstruction and classification.

**Data generation.**   We follow the toy model from Alemi et al. (2018), but add an additional classification label in order to explore supervised and semi-supervised objectives. The true data generating distribution is as follows. We first sample a latent binary variable, $z \sim \mathrm{Ber}(0.7)$, then sample a latent 1D continuous value from that variable, $h|z \sim \mathcal{N}(h|\mu_z, \sigma_z)$, and finally we observe a discretized value, $x = \mathrm{discretize}(h; \mathcal{B})$, where $\mathcal{B}$ is a set of 30 equally spaced bins, and a discrete label, $y = z$ (so the true label is the latent variable that generated $x$). We set $\mu_z$ and $\sigma_z$ such that $R^* \equiv I(x; z) = 0.5$ nats, in the true generative process, representing the ideal rate target for a latent variable model. For the finite dataset, we select 50 examples randomly from the joint $p(x, y, z)$. For the infinite dataset, we directly supply the true full marginal $p(x, y)$ at each iteration during training. When training on the finite dataset, we evaluate model performance against the infinite dataset so that there is no error in the evaluation metrics due to a finite test set.

**Model details.**   We choose to use a discrete latent representation with $K = 30$ values, with an encoder of the form $q(z_i|x_j) \propto -\exp[(w_i^e x_j - b_i^e)^2]$, where $z$ is the one-hot encoding of the latent categorical variable, and $x$ is the one-hot encoding of the observed categorical variable. We use a decoder of the same form, but with different parameters: $q(x_j|z_i) \propto -\exp[(w_i^d x_j - b_i^d)^2]$. We use a classifier of the same form as well: $q(y_j|z_i) \propto -\exp[(w_i^c y_j - b_i^c)^2]$. Finally, we use a variational marginal, $q(z_i) = \pi_i$. Given this, the true joint distribution has the form $p(x, y, z) = p(x)p(z|x)p(y|x)$, with marginal $p(z) = \sum_x p(x, z)$, and conditionals $p(x|z) = p(x, z)/p(z)$ and $p(y|z) = p(y, z)/p(z)$.

The encoder is additionally parameterized following Achille & Soatto (2017) by $\alpha$, a set of learned parameters for a Log Normal distribution of the form $\log \mathcal{N}(-\alpha_i/2, \alpha_i)$. In total, the model has 184 parameters: 60 weights and biases in the encoder and decoder, 4 weights and biases in the classifier, 30 weights in the marginal, and an additional 30 weights for the $\alpha_i$ parameterizing the stochastic encoder. We initialize the weights so that when $\sigma = 0$, there is no noticeable effect on the encoder during training or testing.

**Experiments.**   In Figure 5, we show the optimal, hand-crafted model for the toy dataset, as well as a selection of parameterizations of the TherML objective that correspond to commonly-used objective functions and a few new objective functions not previously described. In the captions, the parameters are specified with $\gamma, \delta, \sigma$ as in the main text, as well as $\rho$, which is a corresponding Lagrange multiplier for $R$, in order to simplify the parameterization. It just parameterizes the optimal surface slightly differently. We train all objectives for 10,000 gradient steps. For all of the objectives described, the model has converged, or come close to convergence, by that point.

Because the model is sufficiently powerful to memorize the dataset, most of the objectives are very susceptible to overfitting. Only the objective variants that are "regularized" by the $S$ term (parameterized by $\sigma$) are able to avoid overfitting in the decoder and classifier.

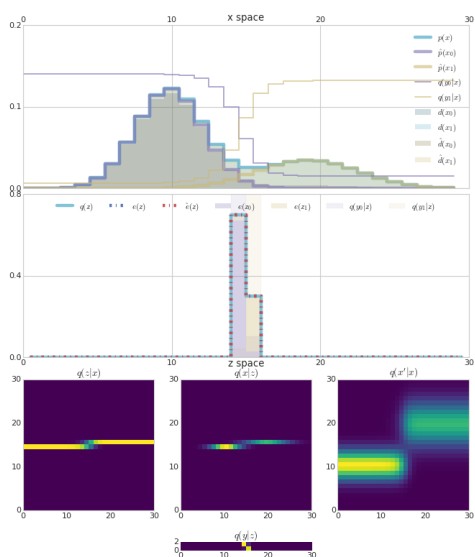

Figure 5: **Hand-crafted optimal model.** Toy Model illustrating the difference between selected points on the three dimensional optimal surface defined by $\gamma$, $\delta$, and $\sigma$. See Section 3 for more description of the objectives, and Appendix H for details on the experiment setup. **Top (i):** Three distributions in data space: the true data distribution, $p(x)$, the model's generative distribution, $g(x) = \sum_z q(z)q(x|z)$, and the empirical data reconstruction distribution, $d(x) = \sum_{x'} \sum_z p(x')q(z|x')q(x|z)$. **Middle (ii):** Four distributions in latent space: the learned (or computed) marginal $q(z)$, the empirical induced marginal $e(z) = \sum_x p(x)q(z|x)$, the empirical distribution over $z$ values for data vectors in the set $\mathcal{X}_0 = \{x_n : z_n = 0\}$, which we denote by $e(z_0)$ in purple, and the empirical distribution over $z$ values for data vectors in the set $\mathcal{X}_1 = \{x_n : z_n = 1\}$, which we denote by $e(z_1)$ in yellow. **Bottom:** Three $K \times K$ distributions: (iii) $q(z|x)$, (iv) $q(x|z)$ and (v) $q(x'|x) = \sum_z q(z|x)q(x'|z)$.

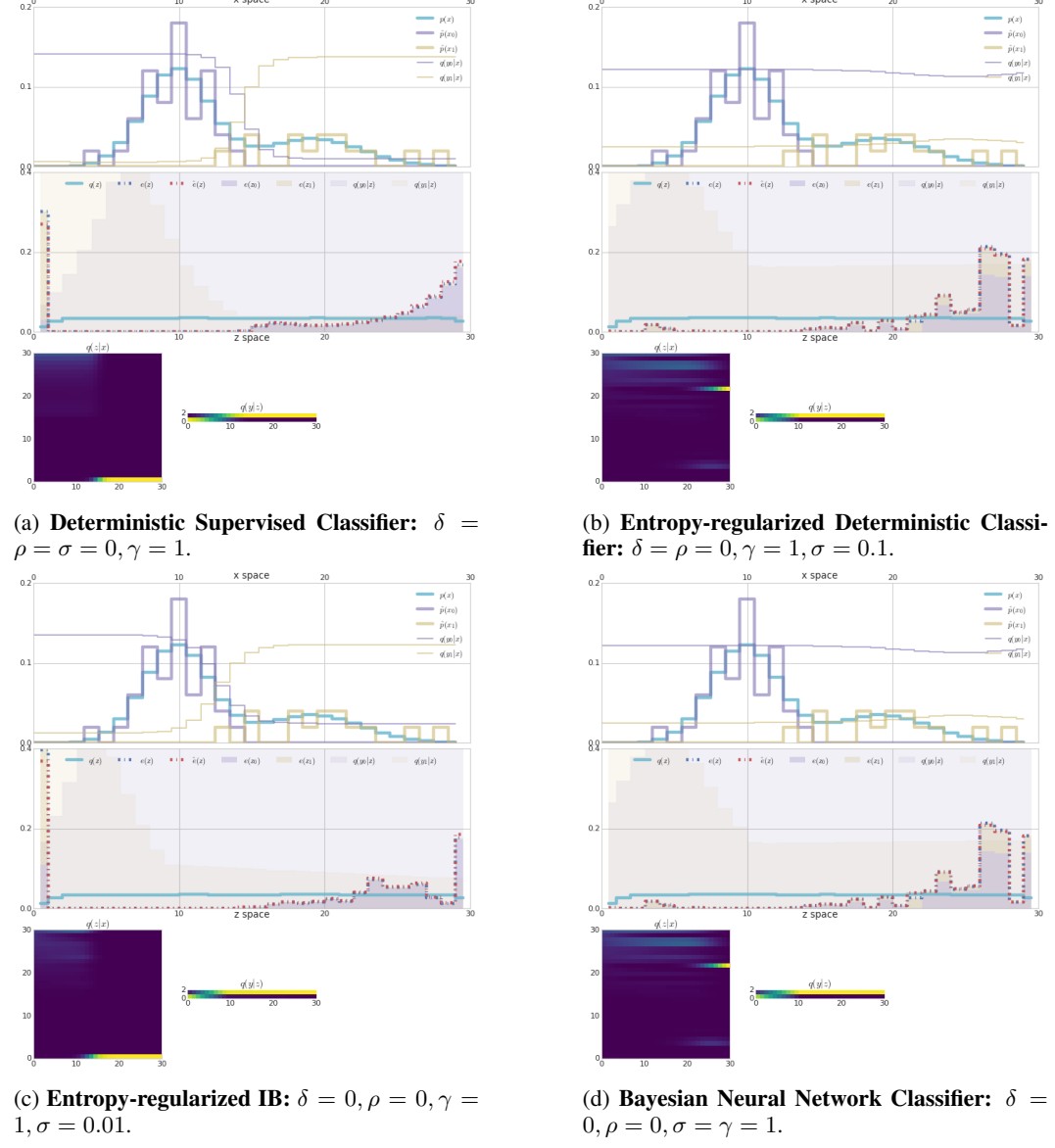

(a) **Deterministic Supervised Classifier:** $\delta = \rho = \sigma = 0, \gamma = 1$.

(b) **Entropy-regularized Deterministic Classifier:** $\delta = \rho = 0, \gamma = 1, \sigma = 0.1$.

(c) **Entropy-regularized IB:** $\delta = 0, \rho = 0, \gamma = 1, \sigma = 0.01$.

(d) **Bayesian Neural Network Classifier:** $\delta = 0, \rho = 0, \sigma = \gamma = 1$.

Figure 6: Supervised Learning approaches.

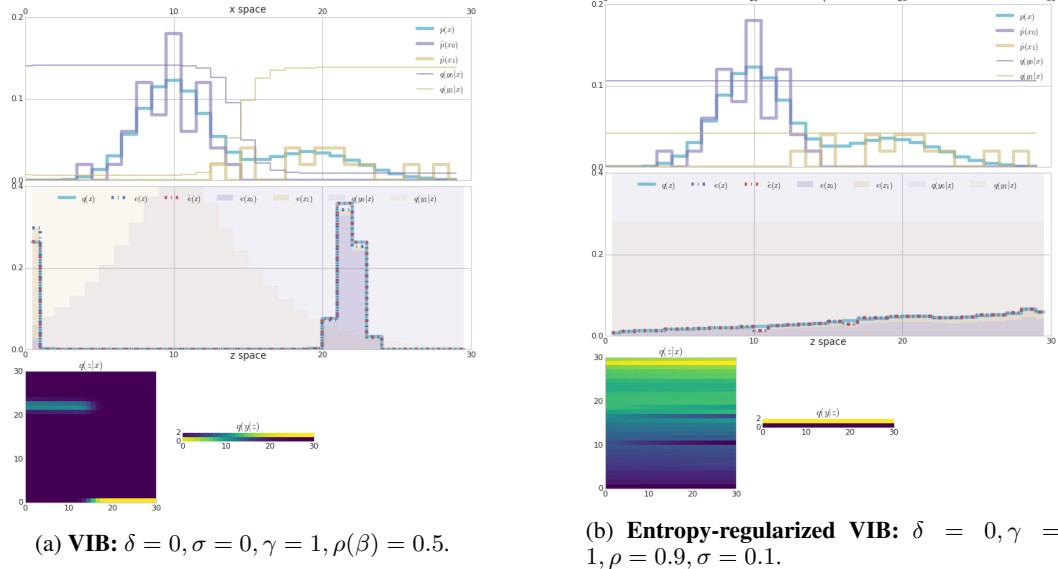

(a) **VIB:** $\delta = 0, \sigma = 0, \gamma = 1, \rho(\beta) = 0.5$.

(b) **Entropy-regularized VIB:** $\delta = 0, \gamma = 1, \rho = 0.9, \sigma = 0.1$.

Figure 7: VIB style objectives.

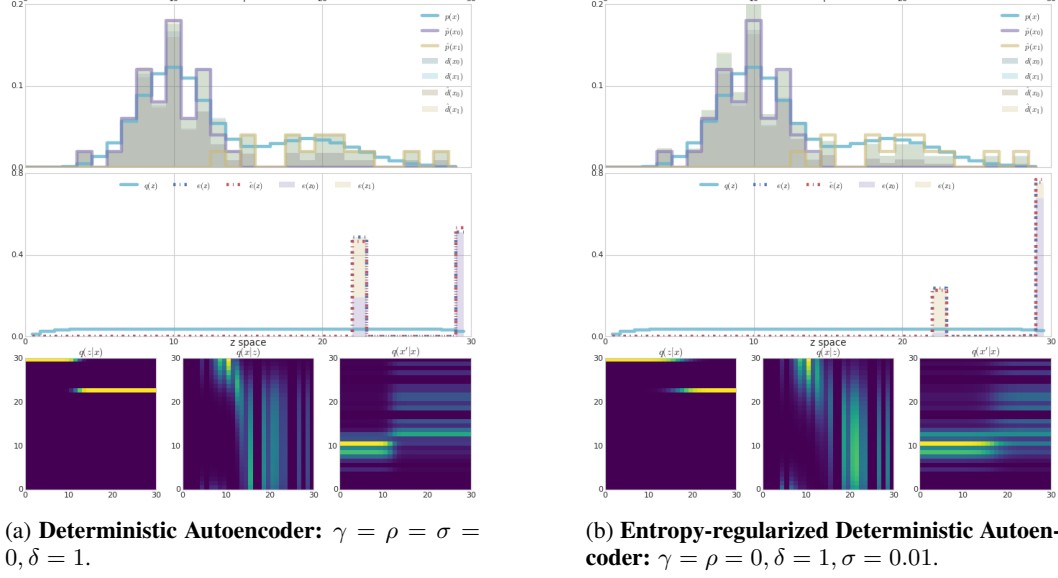

(a) **Deterministic Autoencoder:** $\gamma = \rho = \sigma = 0, \delta = 1$.

(b) **Entropy-regularized Deterministic Autoencoder:** $\gamma = \rho = 0, \delta = 1, \sigma = 0.01$.

Figure 8: Autoencoder objectives.

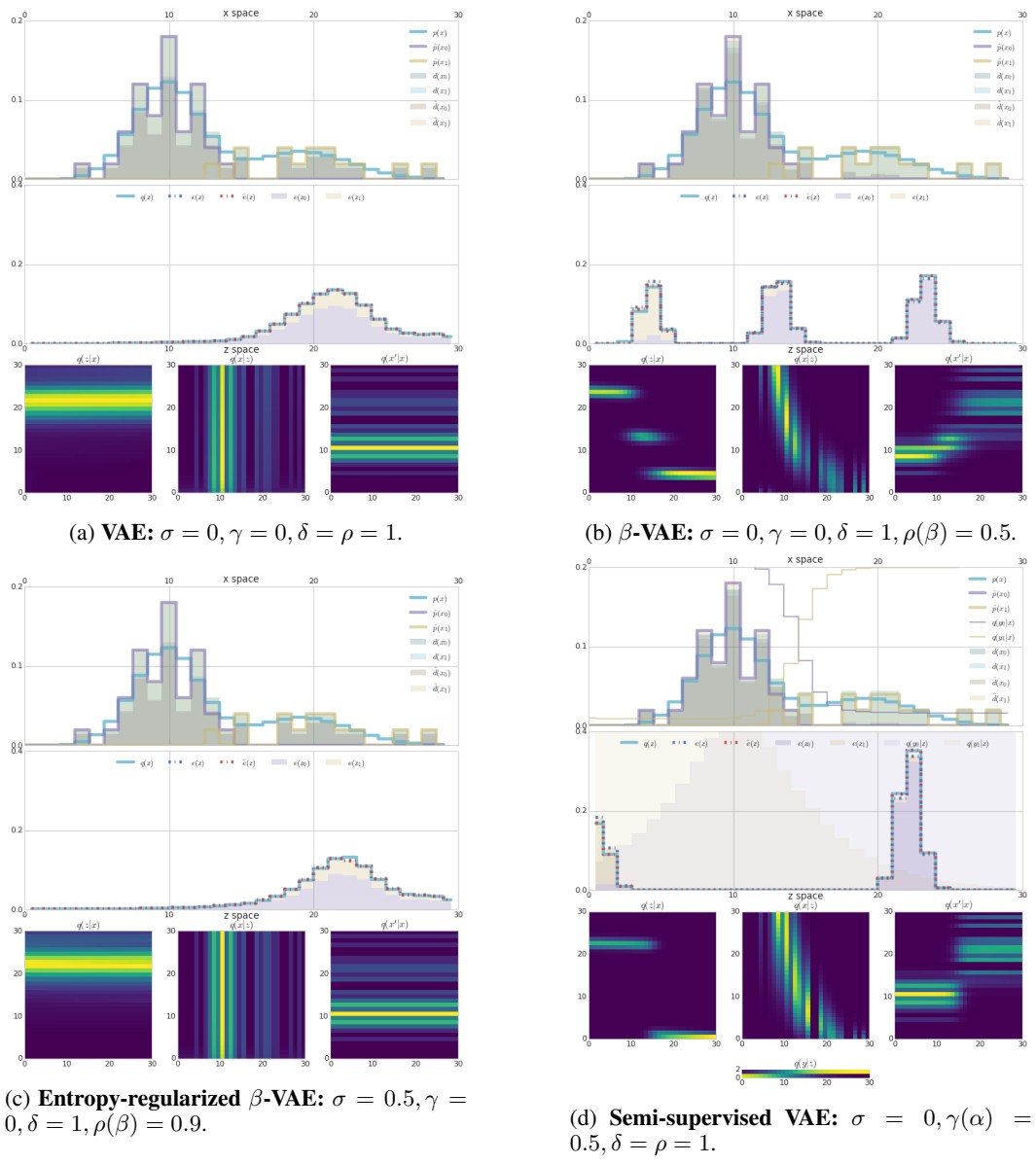

(a) **VAE:** $\sigma = 0, \gamma = 0, \delta = \rho = 1$.

(b) $\beta$**-VAE:** $\sigma = 0, \gamma = 0, \delta = 1, \rho(\beta) = 0.5$.

(c) **Entropy-regularized** $\beta$**-VAE:** $\sigma = 0.5, \gamma = 0, \delta = 1, \rho(\beta) = 0.9$.

(d) **Semi-supervised VAE:** $\sigma = 0, \gamma(\alpha) = 0.5, \delta = \rho = 1$.

Figure 9: VAE style objectives.

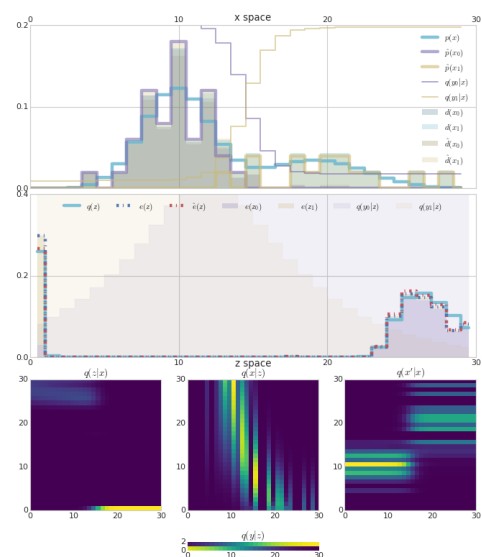

Figure 10: **Full Objective.** $\sigma = 0.5, \gamma = 1000, \delta = 1, \rho = 0.9$. Simple demonstration of the behavior with all terms present in the objective.

