# OpenReview forum: "TherML: The Thermodynamics of Machine Learning"
_ICLR.cc/2019/Conference_

### Official Review · AnonReviewer3 · 2018-11-05
**A formal framework for representation learning.**

**Rating:** 5
**Confidence:** 3

**Review:**

This paper builds on the (Alemi et al 2018) ICML paper and presents a formal framework for representation learning. The authors use a graphical model for their representation learning task and use basic information theoretic inequalities to upper-bound their measure of performance which is a KL divergence. The authors then define the optimal frontier which corresponds to the lowest possible upper-bound and write it as an optimization problem. Written with Lagrange multipliers, they obtain several known cost functions for different particular choices of these parameters.
Then the authors make a parallel with thermodynamics and this part is rather unclear to me. As it is written, this section is not very convincing:
- section 4.1 after equation (27) which function is 'smooth and convex'? please explain why.
- section 4.1 '...the actual content of the law is fairly vacuous...'
- section 4.2 the explanation of equation (30) is completely unclear to me. Please explain better than 'As different as these scenarios appear (why?)...'
- section 4.2 'Just as in thermodynamics, these susceptibilities may offer useful ways to characterize...'
- section 4.2 'We expect...'
- section 4.3 ends with some unexplained equations.
As illustrated by the examples above, the reader is left contemplating this formal analogy with thermodynamics and no hint is provided on how to proceed from here.

---

> ### Author Response · Authors · 2018-11-27
> **Thank you**
>
> Thank you for the review.
>
> We agree Section 4 is rather terse.  Given space constraints we weren't able to describe things in much detail and currently leave too much unsaid.  We thought the analogy was interesting enough to discuss, even if not in detail.
>
> Do you think the paper would be improved if Section 4 was eliminated entirely?  Is the rederivation of existing objectives laid out in the initial sections novel enough to stand on its own?

---

### Official Review · AnonReviewer1 · 2018-11-05
**Formal analogy needs better motivation, clarity, and perhaps worked examples**

**Rating:** 3
**Confidence:** 4

**Review:**

This paper attempts to establish a notion of thermodynamics for machine learning. Let me give an attempt at summary. First, an objective function is established based on demanding that the multi-information of two graphical models be small. The first graphical model is supposed to represent the actual dependence of variables and parameters used to learn a latent description of the training data, and the model demands that the latents entirely explain the correlation of the data, with the parameters marginalized out. Then, a variational approximation is made to four subsets of terms in this objective function, defining four "thermodynamic"  functionals. Minimizing the sum of these functionals puts a variational upper bound on the objective. Next, the sum is related to an unconstrained Lagrange multiplier problem making use of the facts (1) that such an objective will likely have many different realizations of the thermodynamic functionals for specific value of the bound and (2) that on the optimal surface the value of one of the functional can be parametrized in terms of the three others. If we pick the entropy functional to be parameterized in terms of the others, we find ourself precisely in the where the solution to the optimization is a Boltzmann distribution; the coefficients of the Lagrange multipliers will then take on thermodynamic interpretations in of temperature, generalized chemical potentials, etc. At this point, the machinery of thermodynamics can be brought to bear, including a first law, Maxwell relations (equality of mixed partial derivatives), etc.

I think the line of thinking in this paper is very much worth pursuing, but I think this paper requires significant improvement and modifications before it can be published. Part of the problem is that the paper is both very formal and not very clear. It's hard to understand why the authors are establishing this analogy, where they are going with it, what's its use will be, etc. Thermodynamics was developed to explain the results of experiments and is often explained by working out examples analytically on model systems. This paper doesn't really have either such a motivation or such examples, and I think as a result I think it suffers.

I also think the "Tale of Two Worlds" laid out in Section 2 requires more explanation. In particular, I think more can be said about why Q is the the "world we want" and why minimizing the difference between these worlds is the right way to create an objective. (I have no real problem with the objective once it is derived.) Since this paper is really about establishing this formal relationship, and the starting point is supposed to be the motivating factor, I think this needs to be made much clearer.

The I(Z_i, X_i, Theta) - I(X_i, Z_i) terms could have been combined into a conditional mutual information. (I see this is discussed in Appendix A.) This leads to a different set of variational bounds and a different thermodynamics. Why do we prefer one way over the other? At the level of the thermodynamics, what would be the relationship between these different ways of thinking? Since it's hard to see why I want to bother with doing this thermodynamics (a problem which could be assuaged with worked examples or more direct and clear experiments), it's hard to know how to think about this sort of freedom in the analogy. (I also don't understand why the world Q graphical model is different in Appendix A when we combined terms this way, since the world Q lead to the objective, which is independent of how we variationally bound it.) I think ultimately the problem can be traced to the individual terms in the objective (7) not being positive definitive, giving us the freedom to make different bounds by arranging the pieces to get different combinations of positive definite terms. How am I supposed to think about this freedom?

In conclusion, I would really like to see analogies like this worked out and be used to better understand machine learning methods. But for this program to be successful, I think a very compelling case needs to be made for it. Therefore, I think that this paper needs to be significantly rewritten before it can be published.

---

> ### Author Response · Authors · 2018-11-27
> **Thank you**
>
> Thank you for the review and careful read of the paper, and constructive criticism.
>
> We agree that there is more work to be done in further exploring the analogy to thermodynamics, but at least at present thought the existence of the analogy was interesting enough to warrant the current draft.  We hope to develop tighter analogies and analytical examples for simple systems, but realistically we are already very pressed for space.
>
> We agree that more should be said about the choice of world Q and its implications.  Part of the difficulty is the world Q shown in the body of the main work isn't necessarily the best, but it is the one that most directly connects with a wide range of existing objectives in the literature.  We thought it is interesting to note that existing objectives can be motivated as minimizing an information projection to the world Q shown.  We should better emphasize that if the reader finds world Q suspect, but trusts our general program, this could reasonably fuel a suspicion of the utility of the existing objectives.
>
> We are still investigating the utility of the modified objective in Appendix A.  (We note that world Q in Appendix A is Markov equivalent to the one in the main body, the Z<-X arrow simply changed direction.) We suspect it might actually prove a more useful objective in practice than the existing formulation.  We suspect the existing objective has the form it does not for any deep reason but because people naturally think in terms of decoders as a natural element of learning a useful representation.  That in the infinite family limit there is an equivalence in the two forms of objective for a parametric representation with and without a decoder we find interesting.  But at present we can only point out this equivalence as we haven't finished the experimental investigation yet.

---

### Official Review · AnonReviewer2 · 2018-11-08
**Interesting perspective connecting many machine learning objectives**

**Rating:** 7
**Confidence:** 3

**Review:**

This paper introduces an information-theoretic framework that connects a wide range of machine learning objectives, and develops its formal analogy to thermodynamics.
The whole formulation attempts to align graphical models of two worlds P & Q and is expressed as computing the minimum possible relative information (using multi-informations)  between the two worlds. Interestingly, this computation consists of four terms of mutual information, each of which is variationally bounded by a meaningful functional: entropy, rate, classification error, distortion. Finding points on the optimal feasible surface leads to an objective function with the four functionals, and this objective is shown to cover many problems in the literature. The differentials of the objective bring this framework to establish formal analogies between ML and thermodynamics: the first law (the conservation of information), the Maxwell relations, and the second law (relative entropy decrease).

The main contribution of this paper would be to provide a novel and interesting interpretation of previous ML techniques using an objective function in an information theoretic viewpoint. Drawing the objective from the tale of two worlds and connecting them with existing techniques is impressive, and the analogies to thermodynamics are reasonable. I appreciate this new perspective of this paper and think this direction is worth exploring for sure. The terms and relations derived in the course of this work might be useful for understanding or analyzing ML models.

On the other hand, this paper is not easy to follow. It’s written quite densely with technical details omitted, and in some parts lacking proper explanations, contexts, and implications.
E.g.,
- In section 2, why the world Q is what we want?
- Among the mutual information terms, it’s not clear why I(Z_i; X_i, Theta) need to be minimized. After the chain rule, while the part of I(Z_i;Theta | X_i) needs to be minimized, isn’t that I(Z_i; X_i) needs to be maximized?
- The functionals and their roles (Section 2.1) need to be more clarified.
- In the first paragraph of Section 3, why is that “any failure of the distributional families …. feature surface”?
For a broader audience, I recommend the authors to clarify with more explanations, possibly, with motivating examples.
- Formal analogies to thermodynamics (Section 4) are interesting, but remains analogies only without any concrete case of usefulness. The implications of the first and second laws are not explained in detail, and thus I don’t see their significance.  In this sense, section 4 appears incomplete. I hope they are clarified.

---

> ### Author Response · Authors · 2018-11-27
> **Thank you**
>
> Thank you for the review.  We agree the paper is dense at places, but it is already pressing up against the page limit, we were unsure of how to balance the scope of things we wish to talk about with the limited space.
>
> In the original objective (eqn. 7) (that was minimized) I(Z;XΘ) - I(Z;X) appeared, this is identically I(Z;Θ|X) by the chain rule (eqn 9).  Does this clarify why I(Z;Θ|X) is minimized? We will try to make this more clear in the text.
>
> Given space constraints would additional examples in appendices help?
>
> We agree Section 4 is terse at present.  Again, space constraints means it had to be pretty condensed.

---

### Meta-Review · Area_Chair1 · 2018-12-14
**Ambitious aim but not well-enough done**

**Confidence:** 5
**Recommendation:** Reject

**Metareview:**

Connecting different fields and bringing new insights to machine learning are always appreciated. But since it is challenging to do it needs to be done well. This paper falls short here.